# Intrinsic group behaviour II: On the dependence of triad spatial dynamics on social and personal features; and on the effect of social interaction on small group dynamics

**Francesco Zanlungo**[1]*, **Zeynep Yücel**[1,2], **Takayuki Kanda**[1,3]

**1** Intelligent Robotics and Communication Laboratory, ATR, Kyoto, Japan, **2** Department of Computer Science, Okayama University, Okayama, Japan, **3** Department of Social Informatics, Kyoto University, Kyoto, Japan

* zanlungo@atr.jp

**Data Availability Statement:** All relevant data are within the manuscript and its Supporting Information files.

## Abstract

In a follow-up to our work on the dependence of walking dyad dynamics on intrinsic properties of the group, we now analyse how these properties affect groups of three people (triads), taking also in consideration the effect of social interaction on the dynamical properties of the group. We show that there is a strong parallel between triads and dyads. Work-oriented groups are faster and walk at a larger distance between them than leisure-oriented ones, while the latter move in a less ordered way. Such differences are present also when colleagues are contrasted with friends and families; nevertheless the similarity between friend and colleague behaviour is greater than the one between family and colleague behaviour. Male triads walk faster than triads including females, males keep a larger distance than females, and same gender groups are more ordered than mixed ones. Groups including tall people walk faster, while those with elderly or children walk at a slower pace. Groups including children move in a less ordered fashion. Results concerning relation and gender are particularly strong, and we investigated whether they hold also when other properties are kept fixed. While this is clearly true for relation, patterns relating gender often resulted to be diminished. For instance, the velocity difference due to gender is reduced if we compare only triads in the colleague relation. The effects on group dynamics due to intrinsic properties are present regardless of social interaction, but socially interacting groups are found to walk in a more ordered way. This has an opposite effect on the space occupied by non-interacting dyads and triads, since loss of structure makes dyads larger, but causes triads to lose their characteristic V formation and walk in a line (i.e., occupying more space in the direction of movement but less space in the orthogonal one).

**Funding:** FZ is affiliated to Advanced Telecommunications Research Institute International. FZ is supported by JST CREST Program Grant Number JPMJCR17A2, Funder: Japan Society for the Promotion of Science, https://www.jsps.go.jp/english/. ZY is supported by JSPS KAKENHI Grant Number J18K18168, Funder: Japan Society for the Promotion of Science, https://www.jsps.go.jp/english/. ZY is supported by WTT Startup Fund (No grant number available), Funder: Okayama University, http://www.okayama-u.ac.jp/index_e.html. ZY is affiliated to Advanced Telecommunications Research Institute International as visiting researcher to facilitate her collaboration with former colleagues, but does not receive a salary from this company. TK is affiliated to Kyoto University. TK is affiliated to Advanced Telecommunications Research Institute International. TK is supported by JST CREST Program Grant Number JPMJCR17A2, Funder: Japan Society for the Promotion of Science, https://www.jsps.go.jp/english/. The funders had no role in study design, data collection and analysis, decision to publish, or preparation of the manuscript.

**Competing interests:** The authors declare that no competing interests exist. Authors FZ, ZY, and TK are affiliated to a private company, Advanced Telecommunications Research Institute International (ATR), respectively, as part-time contract researcher (FZ), visiting researcher (ZY), and visiting group leader (TK). Nevertheless, their activity in ATR concerning this research work is related to the development of the government and university funded projects stated in the financial disclosure, and not to the development of commercial products or other economical interest of ATR. The authors state that no patent application is going to be pursued based on the research presented on this work, and that the results of this work are not going to be used to promote any kind of economical interest. This does not alter our adherence to PLOS ONE policies on sharing data and materials.

## Introduction

Pedestrian dynamics analysis and simulators often deal with "physical" crowds, i.e. a large number of people located in the same physical area, but not necessarily with a shared social identity (i.e., they are not a "psychological crowd") [1, 2]. Such "physical" crowds present nevertheless a complex social structure if analysed at the microscopic scale, being characterised by the presence of social groups. The number of pedestrians moving in groups depends on the nature of the environment and time of the day [3–5], but it is in general considerable, as groups represent up to 85% of the walking population [6, 7]. Groups have a peculiar dynamics (they move together and close) and as a result not taking into account the presence and behaviour of groups may have an impact on the planning of buildings and of emergency evacuation [8, 9]. A complete assessment of the influence of groups on crowd dynamics is still far from being attained, due also to the lack of quantitative data concerning group behaviour, in particular at medium and high density ranges, and during egress. Nevertheless, a preliminary study combining realistic collision avoidance and state of art group behaviour models [10] shows that groups may have a very strong impact on crowd flow and self-organisation.

For these reasons there is a growing interest in studies concerning group dynamics. Such studies need in general to be based on a "microscopic" approach, (i.e., using models that describe the behaviour of each individual in the crowd, as opposed to models using only macroscopic variables such as crowd density [11]). The microscopic approach allows to cope with differences between individuals, social interactions and psychological aspects [12, 13], but in order to make it possible a quantitative understanding of these differences is needed. More specifically, data concerning how group behaviour changes depending on the nature of the group (e.g. personal features of its components and their social relation) and on the nature of the environment (e.g. crowd density, architectural features, cultural aspects and normal vs egress conditions), have to be collected in order to develop realistic models of group behaviour. Furthermore, as discussed in detail by [14], these data should be collected in *ecological* settings (i.e., observing uninstructed pedestrians in their natural environment). The purpose of this work is to contribute to such a program by providing a quantitative analysis of differences in group behaviour due to the same nature of the group.

In recent years, many works have studied and modelled the specific dynamics of group behaviour [3, 6, 7, 14–33]. For example, [7] assumed that pedestrians in groups tend to walk aligned to facilitate social interaction, while [22] assumed that groups move in an abreast formation when not constrained by the surrounding environment, but may assume a *V* formation or even walk in a line in more congested situations. In [3, 4, 34], we introduced a mathematical model to describe group spatial structure and velocity. [3] introduced a non-Newtonian [35, 36] potential for group interaction, that was able to describe and to some extent predict, in agreement with empirical observations, the *spatial size* (distance between the members), structure and velocity of uninstructed pedestrian groups. In [4] we studied the effect of crowd density (an *extrinsic* property) on group dynamics, and in [34] we proposed a mathematical model to explain the findings of [4] (along with density, other environmental properties such as corridor width may affect group dynamics [37]).

The behaviour of walking groups depends also on its *intrinsic* properties, i.e., group and individual members' features. Age, gender and height are known to affect walking speed (as observed in studies with subjects [38]). Furthermore, group dynamics is expected to be affected also by the relation between the members [14, 21, 39–43]. In particular, [14] suggested gender-related differences in formation and velocity (females walking more abreast and slower than males, and mixed groups walking more abreast than same sex groups). Furthermore, in

recent works, we have shown that it is possible to automatically infer social relation [44] or gender [45] from trajectory data.

In [46] we used a large *ecological* (i.e., obtained by observing uninstructed pedestrians in their natural environment, see [14]) data set and described how spatial structure, spatial size and velocity of *dyads* (two people groups) depend on intrinsic properties of groups, and more specifically on:

1. **purpose** of movement

2. **relation** between the members

3. **gender** of the members

4. **age** of the members

5. **height** of the members

(gender, age and height are, properly speaking, properties of the group members more than of the group itself; in the following, these terms may be considered as shorthands for gender *composition* and similar). Since the data set was based on trajectories of uninstructed pedestrians, all features excluding height (which is automatically provided by the tracking system [47]) are *apparent*, i.e. based on the judgement of human coders. Furthermore, the results are probably influenced by the venue in which data were collected (Osaka, Japan; refer to [48, 49] for an analysis on cultural dependence of pedestrian behaviour). Nevertheless, they provided a useful and quantitative insight into how intrinsic features affect dyadic behaviour. In particular, we observed that relation affects group structure, with couples walking at a very short distance and abreast, colleagues walking less close, and families walking less abreast than friends. Velocity and abreast distance are affected also by pedestrian height, and specifically both velocity and abreast distance grow with average group height. Elderly people walk slowly, while active age adults are faster. Groups with children tend to walk in a non-abreast formation, with a large distance (despite a low abreast distance). A cross-analysis of the interplay between these features (taking into account also the effect of crowd density), confirmed these trends but revealed also a richer structure. For example, the velocity of groups with children appears to increase with density, at least up to the moderate densities presented by the observed environment.

The used data-set was not limited to dyads, but included groups of any size. Anyway, since in the observed environment the number of groups decreases strongly with group size [4], (an effect possibly also due to the difficulty of identifying social links between larger groups of people), the reduced number of triads did not allow us to derive convincing results on their behaviour (preliminary results based on this reduced data set were presented in [50]. For this purpose, we extended our data set asking to a coder to analyse specifically the composition of all triads that had been observed in the large data-set presented by [4]. This new data set (i.e., the explicit intrinsic properties annotation, according to the process described below in Data set, of the triads identified in the data-set of [4]) is the basis of the present analysis of dependence of triad dynamics on intrinsic features.

Both our previous work [46] and the current one are based on the group data set of [4]. As explained below (Data set) this data set consists of annotations of social groups moving in a pedestrian facility. The definition of "social groups" [51] corresponds to people that are not just moving together towards a common goal, but to people who are walking together on purpose. Such groups may have a long lasting social relation between them (e.g. being relatives or colleagues), which usually predates and outlasts their common displacement. A more stringent definition is that of *socially interacting group*, i.e. a group that it is explicitly having some form

of social interaction (usually a conversation) while walking. Socially interacting groups are obviously easier to identify for an external observer, but do not include all social groups, since a portion of these may still be walking together without having an explicit interaction at the moment of observation. Nevertheless, a human observer may use different visual clues to *at least guess* the social relation between the pedestrians (clothing, age, motion patterns, contact, gaze; for example an adult and a child moving together may be guessed to be a parent-child dyad even in absence of explicit social interaction at the moment of observation).

The coder of [4] was asked to explicitly annotate the (apparent) presence of social interaction, and thus this information was present both in the dyad set used in [46] and in the set used for the current work. More in detail, the coder was asked to annotate if two pedestrians were talking to each other, had physical contact, exchanged gaze or clearly watched in the same direction S1 Annex. In [4], having access to a quite large data set, we limited ourselves to the analysis of interacting groups, but in the analysis of [46] we used also non interacting groups. In this work, we make use also of the interaction coding to investigate the effect of social interaction both on dyads and triads (an analysis on "intensity of interaction" was performed, on a reduced data set, using different observables and from a detection-oriented standpoint in [52]).

In triads, it is possible that only two of the pedestrians are socially interacting. Following [3] we assume that the triad structure is determined by the *full interaction of all members*; for this reason, a simpler comparison to the dyadic case, and in order not to reduce too much the non-interacting data sample, *we consider a triad in which only two members are socially interacting as non-interacting.*

## Materials and methods

### Data set

By using 3D range sensors and the algorithm introduced in [47] (which provides, along with the pedestrian position on the ground plane, the height of their head), our research group has collected a very large pedestrian trajectory data set [5]. The data set consists of trajectories automatically tracked in a $\approx 900$ m$^2$ area of the Asia and Pacific Trade Center (ATC, located in the Osaka port area, in Japan) for more than 800 hours during a one year time span. ATC is a multi-purpose building, directly connected to a metro station and including offices, shops, exposition areas and a ferry terminal. Along with the 3D sensor tracking, we video recorded the tracking area using 16 different cameras, and a subset of the video recordings were used by a human coder to identify pedestrian social groups. These labelled group trajectories were used to build the openly available set [53], introduced by [4]. This work is based on a further labelling (of group relation and composition) of this latter data set (although for the purpose of this work we restrict ourselves to data from the corridor area defined in [4], in order to avoid effects due to architectural features of the environment, such as corridor width [37]).

**An ecological data set.** The data set concerns the natural behaviour of pedestrians, i.e. they were behaving in an uninstructed way, and observed in their natural environment (with the consent of local authorities and building managers -approved by the Advanced Telecommunications Research Institute International ethics board, document 502-1-; posters explaining that an experiment concerning pedestrian tracking was being hold were present in the environment.). Although presenting some technical problems such as higher tracking noise, collecting data in the pedestrians' natural environment is becoming more popular [54, 55], as it allows avoiding non-natural behaviour of pedestrians in experiments with subjects (due to the influence of artificial environments, selection of subjects, experimenters' instructions). The relevance of such "artificial" behaviour depends obviously on the purpose of the study, but we

believe that social pedestrian group behaviour could hardly be observed in controlled laboratory experiments [14].

Since we were not able to directly contact the observed pedestrian groups, our approach only provides us the *apparent* belonging to a group and its *apparent* intrinsic properties, as determined by human observers (see the discussion below on coder agreement). This approach has obvious limitations, but we believe that it still provides useful quantitative information about the natural behaviour of actual social groups. Considering the positive response to our work on dyads, and while waiting for the possible development or introduction of more powerful observation methods without disregarding the contribution of experiments performed in more controlled settings, we believe that this work may provide a useful step in the understanding of group dynamics.

**Group composition coding.** In order to obtain the "ground truth" for the inter-group composition and social relation, we proceeded similarly to our previous works [3, 4, 46], and asked coders to observe the video recordings corresponding to triads in the data set [53] and to label, when possible,

1. The apparent purpose of the group's visit to the area (**work** or **leisure**)

2. The apparent gender of their members

3. Their apparent relation (**colleagues**, **friends** or **family**. **Couples**, present in the dyad set, are absent in triads. This categorisation of social relations is based on the work of [56]).

4. Their apparent age (in decades, such as 10-19, etc.)

In [46], we used three different coders, one of them examining the whole data set, while the other two examined only a sub-set of trajectories. In the current work, we used a new coder and for inter-coder agreement analysis we compared to the triad labelling performed by the main coder of [46]. The coders did not have access to our quantitative measurements of position and velocities, and relied only on visual features (e.g., clothing, gestures, behaviour and gazing [57–59]) to identify the social relation and composition (coders had obviously access to visual clues concerning distance and velocity, but not to quantitative measurements. No instruction such as "friends walk closer than colleagues" or similar was provided to them, since they were simply told to use the available visual clues to code the social relations and composition.). Information regarding instructions to coders, coding criteria and coders' consent may be found in S1 Annex.

Although pedestrian ages were intended to be coded "in decades" (e.g., 10-19 years, etc.), for children with an apparent age below 10 year old the coder assigned an explicit number from 0 to 9. It is possible that if the coder had used only the 0-9 and 10-19 years categories, some of the children coded with an age close but lower than 10 could have been put in the 10-19 category. When performing the data analysis we thus decided to use the 0-7 and 8-19 categories to group children age, as such categories seemed to be less ambiguous.

Any sort of inconsistency in the coding (e.g., when a group was composed of mixed relation members, such as a colleague and two friends, or when some entry was left empty by mistake) was dealt with by simply not using the data. We also excluded from our data set all groups that included wheelchairs or strollers (whose presence was coded in the original [4] data-set).

The main coder of this work is a part-time research assistant working in the ATR laboratories, who is not specialised in pedestrian studies and is not aware of our mathematical models of pedestrian behaviour. The main coder of [46] was a short time internship student, again not specialised in pedestrian studies and not aware of our mathematical models of pedestrian behaviour.

**Coders' agreement.**   The coding process obviously depends on the subjective evaluation of the coder. Nevertheless, we could use the 163 triads examined both by the main coder in this work and the main coder in [46] to evaluate the reliability of their coding. To this end, we use in S1 Appendix two different approaches. On one hand, we compare the coding results using statistical indicators such as Cohen's kappa [60] and Krippendorf's alpha [61]. On the other hand, we also treat the different codings as independent experiments, and quantitatively and quantitatively compare the findings. Both approaches suggest that the coding process of all categories, and in particular of purpose, relation and gender, is highly reliable.

**Trajectories.**   Pedestrian positions and velocities are tracked by the system of [47] at $\delta t$ in the order of tens of milliseconds, and then, in order to reduce tracking noise and the influence of gait, averaged over $\Delta t = 0.5$ s. At the end of this process, pedestrian positions are given at discrete times $k$, as

$$\mathbf{x}(k\Delta t) = (x(k\Delta t), y(k\Delta t), z(k\Delta t)), \tag{1}$$

where $z$ gives the height of the top of the pedestrian head. Velocities are then defined (in 2D) by using

$$\mathbf{v}(k\Delta t) = [(x(k\Delta t) - x((k-1)\Delta t))/\Delta t, (y(k\Delta t) - y((k-1)\Delta t))/\Delta t]. \tag{2}$$

**Pedestrian height.**   In order to avoid the instabilities in the measurement of pedestrians' -head- height described in [47], we consider the average of $z$ measurements larger than the median (as computed over the entire observation period). In addition, so as to cope with ID swaps (between different pedestrians or between a pedestrian and an object), we make use of the fact that group members always stay in a reasonable proximity along their locomotion. Specifically, considering the trajectory points which are within a distance of 4 m to each other (a threshold determined based on our findings in [3, 4]), we ensure that only the pedestrians who move as a member of the group are considered.

Following [3, 4], only data points where both the average group velocity $V$ (Eq 3) and all individual velocities $v_i$ are larger than 0.5 m/s, and with all pedestrian positions falling inside a square whose centre corresponds to the geometrical one of the group, were used. The square has sides of $L = 2.5$ meters for dyads, and $L = 3$ meters for triads (all these thresholds were again based on our analysis of probability distribution functions of group positions in [3, 4] and pedestrian velocities in [5, 62].).

**Density.**   Obviously, velocity and spatial configuration of pedestrian groups are not independent of crowd density [4]. In order to address the effect of crowd density on group dynamics, we use the specifics of the approach of [4], and favour spatial resolution over temporal resolution in the computation of density. Namely, we compute empirical values of pedestrian density in fairly small cells (i.e., $L = 0.5$ meters square cells on the 2D plane) and in somewhat long time intervals (300 seconds). Further information on the details of this procedure can be found in [4], while several alternative density computation methods may be found in [63–65].

## Quantitative observables

Following [3, 4, 34], in [46] we defined the following quantitative observables for the dynamics of dyads (Fig 1):

1. *Group velocity V*, given by

$$V = |\mathbf{V}|, \qquad \mathbf{V} = (\mathbf{v}_1 + \mathbf{v}_2)/2, \tag{3}$$

   $\mathbf{v}_i$ being the velocities of the two pedestrians in an arbitrary reference frame co-moving

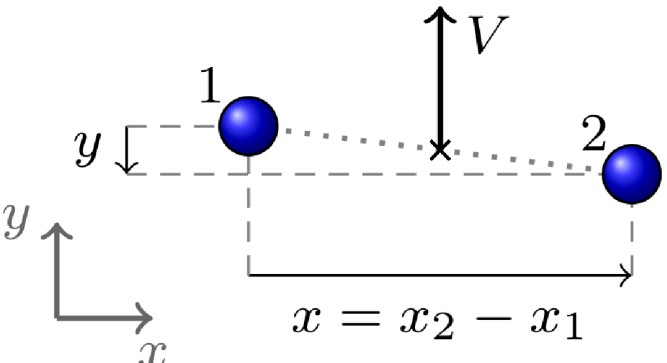

**Fig 1. Group observables in dyads.**

with the environment (i.e. in which the velocity of walls and other architectural features is zero).

2. *Pedestrian distance* or *group spatial size r*, given by

$$r = |\mathbf{r}| \qquad \mathbf{r} = \mathbf{r}_1 - \mathbf{r}_2, \tag{4}$$

$\mathbf{r}_i$ being the positions of the two pedestrians in the above reference frame.

3. *Group abreast distance* or *abreast extension* or *group depth x*, defined as follows. We first define the group velocity unit vector (versor)

$$\hat{\mathbf{g}} = \frac{\mathbf{V}}{V}. \tag{5}$$

Then, for each pedestrian we compute the clockwise angle $\theta_i$ between $\hat{\mathbf{g}}$ and $\mathbf{r}_i$, and define the projection of each $\mathbf{r}_i$ orthogonal to the velocity as

$$x_i = r_i \sin \theta_i. \tag{6}$$

If necessary, we reassign the pedestrian labels to obtain $x_1 \leq x_2$ and finally define the abreast distance as

$$x = x_2 - x_1 \geq 0. \tag{7}$$

4. The *group extension in the direction of motion*, or *group depth*, is, as suggested by the name, the spatial size of the group in the direction of the group velocity, or

$$y_i = r_i \cos \theta_i, y = |y_2 - y_1|. \tag{8}$$

In a similar way, for triads (Fig 2) we define the group velocity as

$$\mathbf{V} = \sum_i \mathbf{v}_i / 3 \tag{9}$$

and use a reference frame whose *y* axis is aligned with $\mathbf{V}$ to measure the components $(x_i, y_i)$ of all pedestrians. The labels are chosen in such a way that $x_1 \leq x_2 \leq x_3$. Our quantitative analysis will be based on the following observables:

1. *Group velocity*, defined as

$$V = |\mathbf{V}| \tag{10}$$

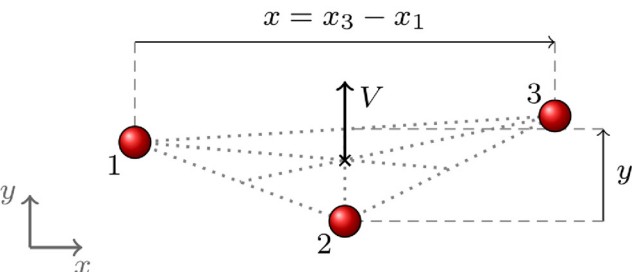

**Fig 2. Group observables in triads.**

2. *Group width*, defined as

$$x = x_3 - x_1 \qquad (11)$$

3. *Group depth*, defined as

$$y = |y_3 + y_1 - 2y_2|/2 \qquad (12)$$

4. For triads, the definition of *r* is less immediate. To measure properly the spatial size of the group we have decided to define first the centre of mass position as

$$\mathbf{X} = \sum_i \mathbf{r}_i/3, \qquad (13)$$

the distance from the centre of mass of each pedestrian as

$$r_i^{\mathrm{cm}} = |\mathbf{r}_i - \mathbf{X}|, \qquad (14)$$

and finally the *group spatial size* as

$$r = \sum_i r_i^{\mathrm{cm}}/3. \qquad (15)$$

In what follows, for each observable (i.e. *V*, *r*, *x* and *y*) and intrinsic factor (i.e., purpose, relation, gender, age and height), we present four values as the number of groups $N_g^k$, the observable average $< O >_k$ (where *O* is a generic symbol standing for one of the observables), standard deviation $\sigma_k$ and standard error $\varepsilon_k$, (see S2 Appendix for details) reported in the form

$$< O >_k \pm \varepsilon_k \ \ (\sigma). \qquad (16)$$

In addition, so as to assess the variations regarding different categories of each intrinsic factor, we present ANOVA *F* function and *p*-values, effect size $\delta$, and coefficient of determination *R* (see S2 Appendix for a detailed definition of these indicators). Results of a detailed analysis, where the effect of each variable is cross-referenced across different intrinsic factors, are presented in S3 Appendix.

## Results

### The effect of purpose

**Overall statistical analysis.** The purpose dependence of all observables for the 687 triads that provided enough data points to be analysed and whose purpose was coded are shown in Table 1 (refer to S2 Appendix for an explanation of all terms). Results concerning dyads (from [46]) are reported in Table 2.

**Table 1. Observable dependence on purpose for triads.** For each observable we provide the number of groups $N_g^k$, the average, standard error, standard deviation (in parenthesis), ANOVA $F$ function and $p$-values, effect size $\delta$, and coefficient of determination $R$. Lengths in millimetres, times in seconds.

| Purpose | $N_g^k$ | $V$ | $r$ | $x$ | $y$ |
|---|---|---|---|---|---|
| Leisure | 608 | 1005 ± 6.4 (157) | 602 ± 5.9 (145) | 909 ± 11 (267) | 704 ± 15 (364) |
| Work | 79 | 1194 ± 18 (159) | 619 ± 17 (151) | 1111 ± 31 (271) | 631 ± 42 (372) |
| $F_{1,685}$ | | 101 | 0.937 | 40 | 2.75 |
| $p$ | | $<10^{-8}$ | 0.333 | $<10^{-8}$ | 0.0977 |
| $R^2$ | | 0.129 | 0.00137 | 0.0551 | 0.004 |
| $\delta$ | | 1.2 | 0.116 | 0.757 | 0.199 |

A comparison between the dyadic and triadic results shows, first of all, that we have a strong lack of balance between the number of triads observed in each category, which partially hinders our analysis. Nevertheless, the amount of data is large enough to verify that, as for dyads, work-oriented triads are faster in a statistically significant way. Leisure oriented triads also walk closer and with lower abreast distance than work-oriented ones (again in agreement with dyads) but with a larger group depth (in agreement with dyads). Although the result concerning $r$ and $y$ are not statistically significant, it should be noted that the effect sizes (independent from sample size) are comparable between dyads and triads. Finally, in agreement with [3, 4, 34], triads are slower than dyads, the effect being stronger in leisure oriented groups. As can be noticed by the results of the following sections, triads are always slower than dyads for any value of all intrinsic or extrinsic properties. We recall that according to the mathematical formulation of [3], the lower velocity of groups with respect to pedestrian walking individually is due to the non-Newtonian interaction terms. Furthermore, since the number of such interaction terms grows faster than group size (assuming just first-neighbour interactions we have 2 $(n − 1)$ ordered pairs in a $n$ people groups), velocity is a decreasing function of groups size. Although in the original model of [3] the non-Newtonian term was introduced as a tendency to keep the partners in one's field of view, which could also be considered as a coordination strategy, a proposed alternative explanation of the non-Newtonian term was that it could express the "cognitive load" of social interaction.

**Probability distribution functions.** By studying the probability distribution functions for the observables $V$, $r$, $x$ and $y$, shown respectively in Figs 3, 4, 5 and 6, and whose statistical analysis is reported in S4 Appendix (refer again to S2 Appendix for the difference between the analysis reported in the main text and the one of S4 Appendix), we can better understand the differences in behaviour between workers and leisure oriented people.

We may observe that the $x$ and $V$ peaks and tails are displaced to higher values for workers. The $r$ peak is also displaced to a higher value, although the tails of the distributions are very similar. Correspondingly, the $y$ distribution is slightly more spread in leisure-oriented

**Table 2. Observable dependence on purpose for dyads.** Lengths in millimetres, times in seconds.

| Purpose | $N_g^k$ | $V$ | $r$ | $x$ | $y$ |
|---|---|---|---|---|---|
| Leisure | 716 | 1118 ± 7.3 (195) | 815 ± 9.5 (253) | 628 ± 6.1 (162) | 383 ± 12 (334) |
| Work | 372 | 1271 ± 8.2 (158) | 845 ± 12 (228) | 713 ± 8 (154) | 332 ± 15 (289) |
| $F_{1,1086}$ | | 169 | 3.75 | 69.4 | 6.25 |
| $p$ | | $<10^{-8}$ | 0.053 | $<10^{-8}$ | 0.0126 |
| $R^2$ | | 0.135 | 0.00344 | 0.0601 | 0.00572 |
| $\delta$ | | 0.832 | 0.124 | 0.533 | 0.16 |

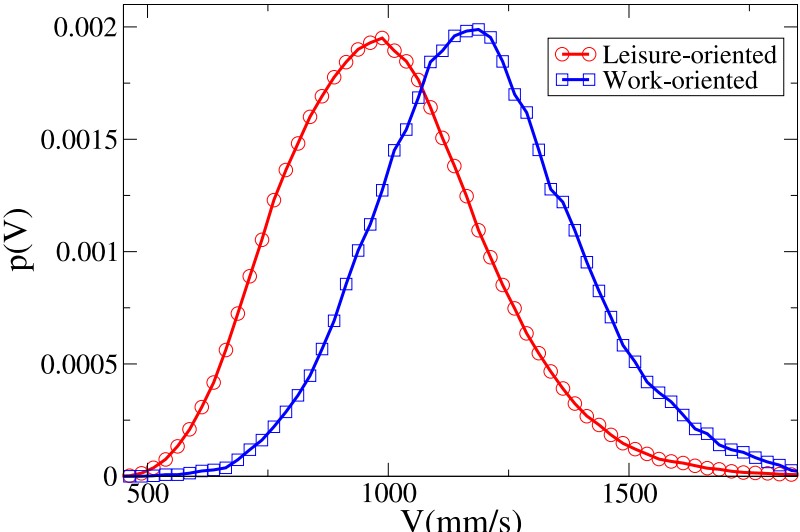

**Fig 3. Pdf of the *V* observable in triads according to purpose.** Red, centre of bin identified by circles: leisure-oriented. Blue, centre of bin identified by squares: work-oriented. All pdfs in this work are shown after smoothing with a moving average filter.

pedestrians. These results, suggesting that leisure-oriented groups are less ordered (i.e., less aligned orthogonally to the direction of motion), have a correspondence with those observed in dyads.

## The effect of relation

We also analyse the dependence of group dynamics on the social relation between their members. As expected, pedestrians that are coded as "work oriented", are usually coded into the "colleagues" relation category (and similarly, those coded as "leisure oriented" fall into one of the "families", "friends" or "couples" categories). Nevertheless, there is a clear conceptual

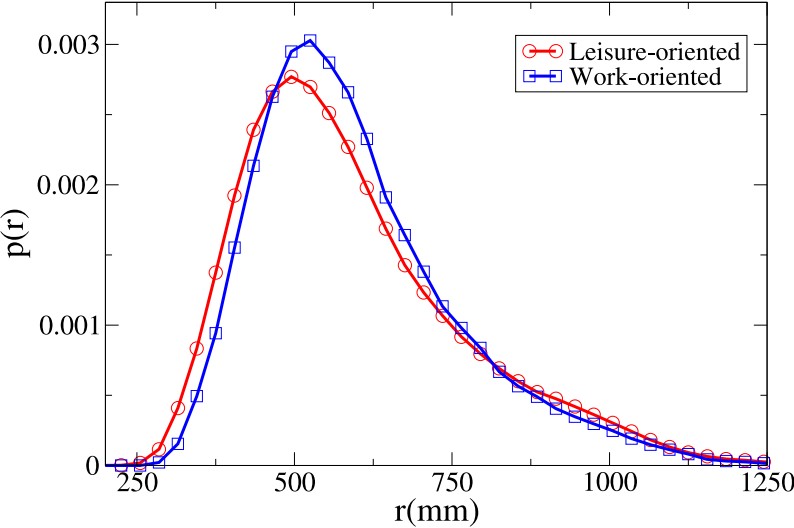

**Fig 4. Pdf of the *r* observable in triads according to purpose.** Red and circles: leisure-oriented. Blue and squares: work-oriented.

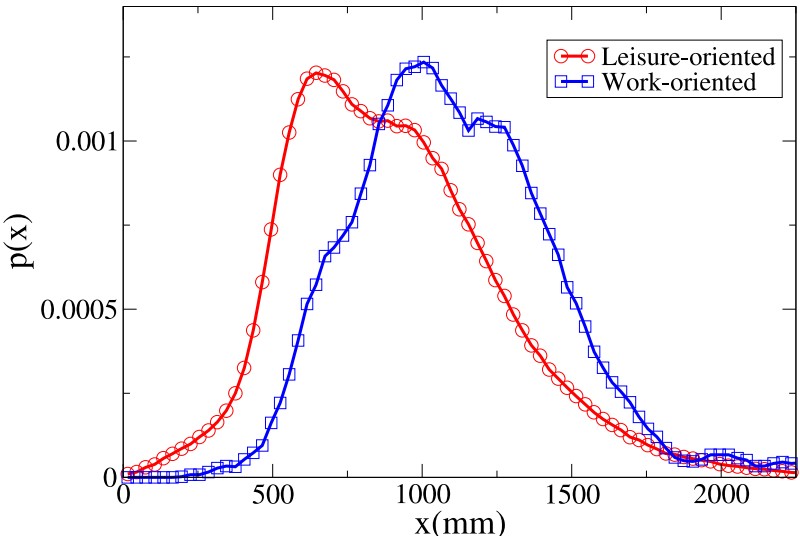

**Fig 5. Pdf of the *x* observable in triads according to purpose.** Red and circles: leisure-oriented. Blue and squares: work-oriented.

difference between the purpose and relation categories (for example, colleagues may go to the shopping mall for lunch or other leisure activities outside working time), and thus we provide an independent analysis. Nevertheless, since in the triad data set the correspondence between colleagues and work-oriented is perfect, in this work we perform cross-analysis only for the relation property, skipping the one based on purpose.

**Overall statistical analysis.**   The relation dependence of all observables for the 687 triads that provided enough data points to be analysed, and whose relation was coded, are shown in Table 3, while the corresponding dyad results from [46] are in Table 4.

As it happens for dyads, also in triads colleagues walk considerably faster than pedestrians in other relation categories. Friends are faster than families, with a difference in the average

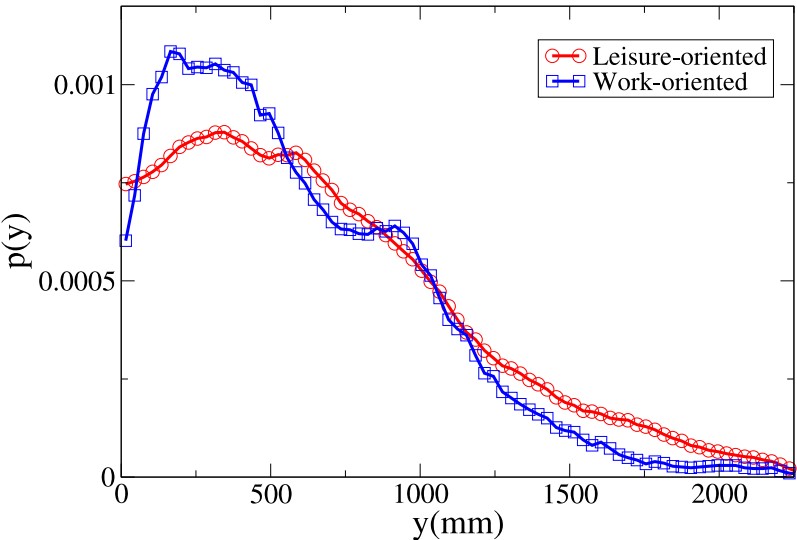

**Fig 6. Pdf of the *y* observable in triads according to purpose.** Red and circles: leisure-oriented. Blue and squares: work-oriented.

**Table 3. Observable dependence on relation for triads.** Lengths in millimetres, times in seconds.

| Relation | $N_g^k$ | V | r | x | y |
|---|---|---|---|---|---|
| Colleagues | 79 | 1191 ± 18 (157) | 624 ± 17 (151) | 1103 ± 31 (277) | 647 ± 42 (377) |
| Families | 396 | 987 ± 7.8 (156) | 622 ± 7.7 (154) | 878 ± 13 (261) | 753 ± 19 (386) |
| Friends | 212 | 1038 ± 11 (156) | 564 ± 8.1 (117) | 970 ± 18 (268) | 605 ± 20 (295) |
| $F_{2,684}$ | | 56.8 | 12.3 | 26.8 | 12.5 |
| p | | $<10^{-8}$ | $5.65 \cdot 10^{-6}$ | $<10^{-8}$ | $4.62 \cdot 10^{-6}$ |
| $R^2$ | | 0.142 | 0.0347 | 0.0727 | 0.0353 |
| $\delta$ | | 1.31 | 0.47 | 0.855 | 0.415 |

**Table 4. Observable dependence on relation for dyads.** Lengths in millimetres, times in seconds.

| Relation | $N_g^k$ | V | r | x | y |
|---|---|---|---|---|---|
| Colleagues | 358 | 1274 ± 8.3 (157) | 851 ± 12 (231) | 718 ± 8.3 (157) | 334 ± 15 (292) |
| Couples | 96 | 1099 ± 17 (169) | 714 ± 22 (219) | 600 ± 15 (150) | 291 ± 24 (231) |
| Families | 246 | 1094 ± 13 (197) | 863 ± 19 (302) | 583 ± 11 (171) | 498 ± 25 (391) |
| Friends | 318 | 1138 ± 11 (200) | 792 ± 11 (199) | 662 ± 7.5 (134) | 314 ± 15 (268) |
| $F_{3,1014}$ | | 60.7 | 12.2 | 42.3 | 21.4 |
| p | | $<10^{-8}$ | $7.39 \cdot 10^{-8}$ | $<10^{-8}$ | $<10^{-8}$ |
| $R^2$ | | 0.152 | 0.0349 | 0.111 | 0.0595 |
| $\delta$ | | 1.03 | 0.529 | 0.828 | 0.587 |

velocity roughly equivalent to 5 standard errors. Colleagues and families keep the largest absolute distance *r*, while friends walk at the closest one. On the other hand, concerning abreast distance *x*, the lowest value is assumed in families, followed by friends and colleagues. Group depth *y* assumes the smallest value in friends and the highest value in families. Accounting for the absence of couples, these results are basically equivalent to those found in dyads.

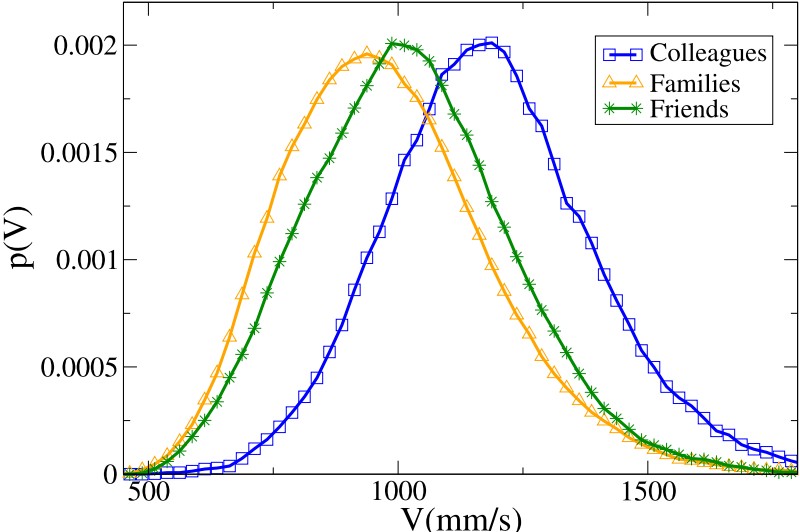

**Fig 7. Pdf of the *V* observable in triads according to relation.** Blue and squares: colleagues. Orange and triangles: family. Green and stars: friends.

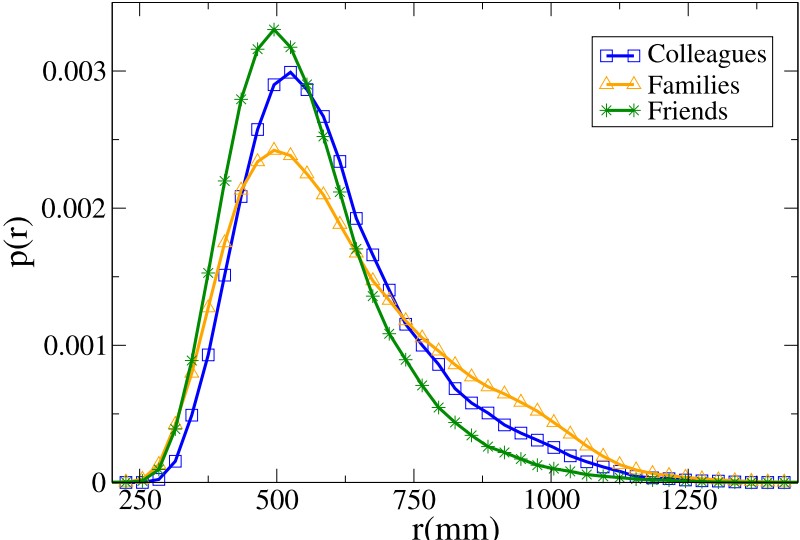

**Fig 8. Pdf of the *r* observable in triads according to relation.** Blue and squares: colleagues. Orange and triangles: family. Green and stars: friends.

**Probability distribution functions.** These results may be completely understood only by analysing the probability distribution functions, which are shown in Figs 7, 8, 9 and 10 for, respectively, *V*, *r*, *x* and *y* (the statistical analysis of these distributions is reported in S4 Appendix).

The *V* distributions for families, friends and colleagues present, in this order, growing peak values and tails displaced on the right. Although the colleague distribution is clearly distinct, also the difference between families and friends is quite evident (stronger than in the dyadic case).

For *r*, the peak position assumes the minimum value in families and friends, but the former distribution presents the fattest tail, a result in agreement with the dyadic one. The

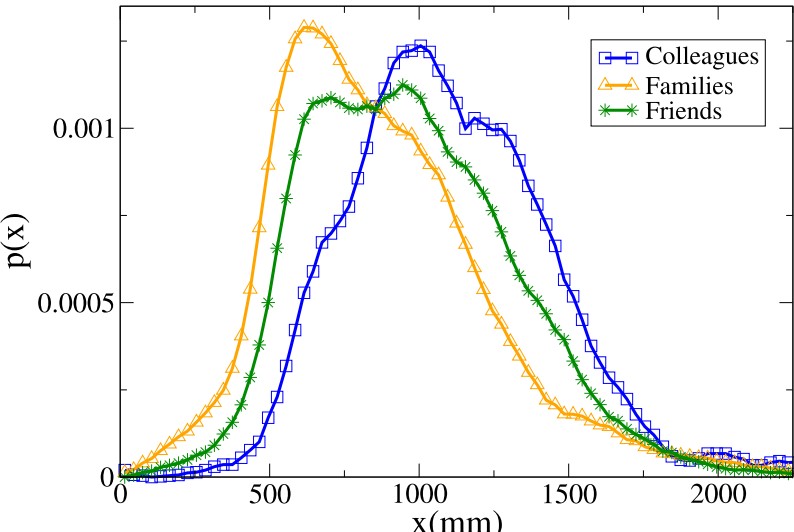

**Fig 9. Pdf of the *x* observable in triads according to relation.** Blue and squares: colleagues. Orange and triangles: family. Green and stars: friends.

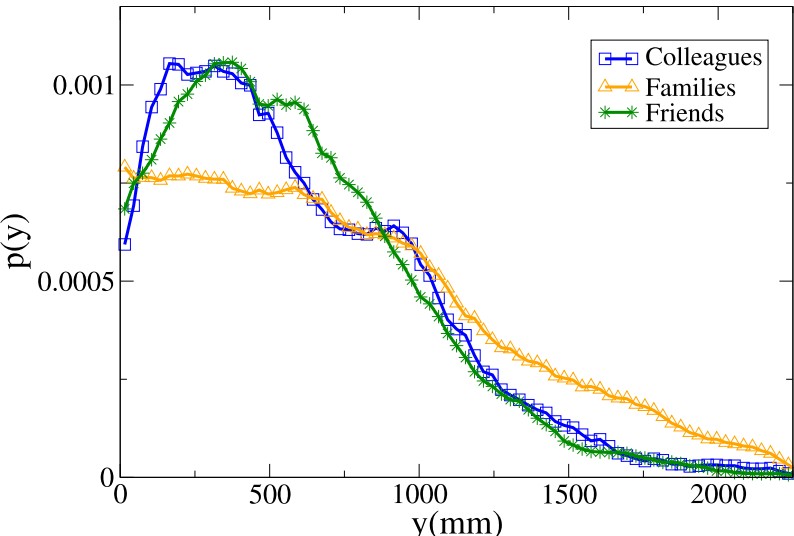

**Fig 10. Pdf of the *y* observable in triads according to relation.** Blue and squares: colleagues. Orange and triangles: family. Green and stars: friends.

distributions of colleagues and friends are similar, although the colleague one is displaced to higher values.

Concerning *x*, friends have a distribution that falls in between the family (assuming lower values) and the colleague one (higher values). The friend distribution presents also a wider peak. Finally, while the *y* distribution is quite similar between colleagues and friends, it is very spread for families (less ordered behaviour). We recall that in [46] it was observed that for dyads the presence of high *y* values in families is, at least partially, explained by the presence of children that may exhibit a more erratic behaviour.

Aside from a bigger difference between the velocity distributions of families and friends (and taking into account the absence of couples), the results are considerably similar to the dyadic ones.

**Further analysis.** In S3 Appendix we analyse how these results depend on the age, gender, density and height of the group members, while in S1 Appendix we verify whether these findings are confirmed by all coders. This analysis confirms all the trends exposed above.

### The effect of gender

**Overall statistical analysis.** The gender dependence of all observables for the 687 triads that provided enough data points to be analysed, and whose gender was coded, are shown in Table 5, while the corresponding dyad data from [46] are shown in Table 6. We may see that the differences between the distributions are statistically significant for each observable. As it had been observed for dyads, from the standpoint of velocity the main difference is between faster all male groups and slower groups that include at least one female. Between the latter, all female ones are slightly faster. From a distance standpoint, the following facts can be stated. There is a clear difference between same-gender and mixed gender groups, the latter having lower abreast distance but larger depth (i.e. a less ordered structure). Between same-gender triads, it can be noticed that males walk at a larger distance than females. The effect sizes, in particular for *r* and *y*, are larger in triads.

**Table 5. Observable dependence on gender for triads.** Lengths in millimetres, times in seconds.

| Gender | $N_g^k$ | $V$ | $r$ | $x$ | $y$ |
|---|---|---|---|---|---|
| Three females | 152 | 1027 ± 12 (147) | 557 ± 9.8 (120) | 968 ± 22 (267) | 597 ± 26 (322) |
| Two females | 231 | 972 ± 9 (137) | 609 ± 9.5 (144) | 872 ± 17 (257) | 732 ± 25 (386) |
| Two males | 173 | 1008 ± 12 (164) | 637 ± 12 (157) | 915 ± 21 (277) | 768 ± 27 (354) |
| Three males | 131 | 1145 ± 17 (189) | 608 ± 13 (147) | 1020 ± 25 (283) | 649 ± 31 (360) |
| $F_{3,683}$ | | 34.5 | 8.63 | 9.48 | 7.69 |
| $p$ | | $<10^{-8}$ | $1.26 \cdot 10^{-5}$ | $3.85 \cdot 10^{-6}$ | $4.64 \cdot 10^{-5}$ |
| $R^2$ | | 0.132 | 0.0365 | 0.04 | 0.0327 |
| $\delta$ | | 1.09 | 0.568 | 0.553 | 0.506 |

**Table 6. Observable dependence on gender for dyads.** Lengths in millimetres, times in seconds.

| Gender | $N_g^k$ | $V$ | $r$ | $x$ | $y$ |
|---|---|---|---|---|---|
| Two females | 252 | 1102 ± 12 (193) | 790 ± 14 (227) | 647 ± 7.8 (123) | 321 ± 20 (311) |
| Mixed | 371 | 1111 ± 9.5 (183) | 824 ± 14 (273) | 613 ± 9 (174) | 416 ± 18 (350) |
| Two males | 466 | 1254 ± 8.3 (178) | 846 ± 11 (228) | 699 ± 7.7 (166) | 349 ± 14 (293) |
| $F_{2,1086}$ | | 84.6 | 4.37 | 30.7 | 7.69 |
| $p$ | | $<10^{-8}$ | 0.0129 | $<10^{-8}$ | 0.000484 |
| $R^2$ | | 0.135 | 0.00798 | 0.0535 | 0.014 |
| $\delta$ | | 0.825 | 0.248 | 0.51 | 0.282 |

**Probability distribution functions.**   Further insight can be obtained analysing the probability distribution functions, which are shown in Figs 11, 12, 13 and 14 for, respectively, the $V$, $r$, $x$ and $y$ observables (the statistical analysis of these distributions is reported in S4 Appendix).

The all-male velocity distribution is clearly displaced to higher values, while female and mixed gender distributions are relatively similar. On the other hand, when examining distance distributions, in particular concerning the $x$ and $y$ variables, we may see that mixed-gender

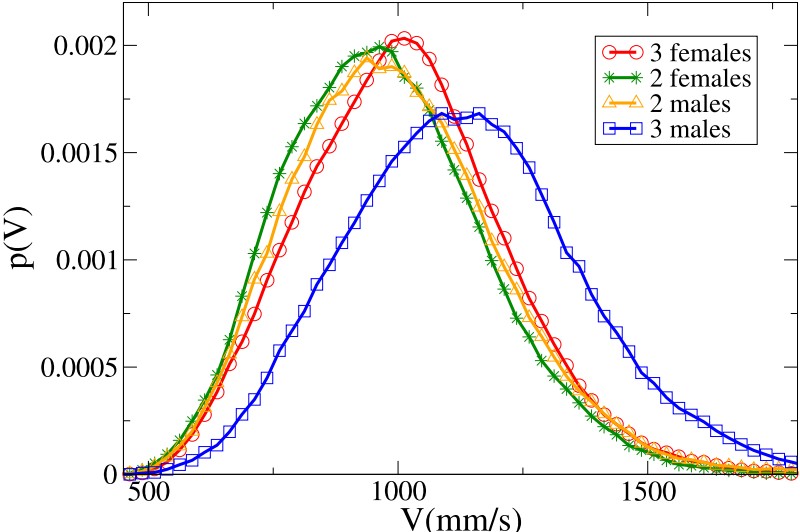

**Fig 11. Pdf of the $V$ observable in triads according to gender.** Red and circles: three females. Green and stars: two females. Orange and triangles: two males. Blue and squares: three males.

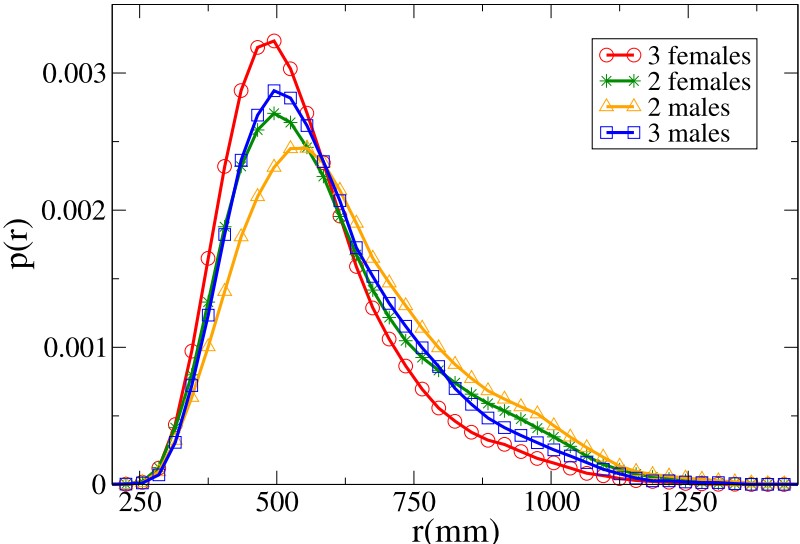

**Fig 12. Pdf of the *r* observable in triads according to gender.** Red and circles: three females. Green and stars: two females. Orange and triangles: two males. Blue and squares: three males.

distributions are qualitatively similar between them, and distinct from the same-gender ones (the two males *r* distribution assumes clearly its peak at a higher value than the two females one). A comparison with Figs 9 and 10 clearly suggests an expectable overlap between families and mixed-gender triads.

**Further analysis.** Further insight is obtained by analysing the interplay of gender with other effects, in particular those related to relation, as shown in S3 Appendix (coder reliability is analysed in S1 Appendix). There, we observe that difference in velocity between males and females is present also when relation is kept fixed, but its effect size is reduced. Furthermore, as expected, the more ordered (lower *y*) behaviour of same gender triads is influenced by the limited overlapping with families.

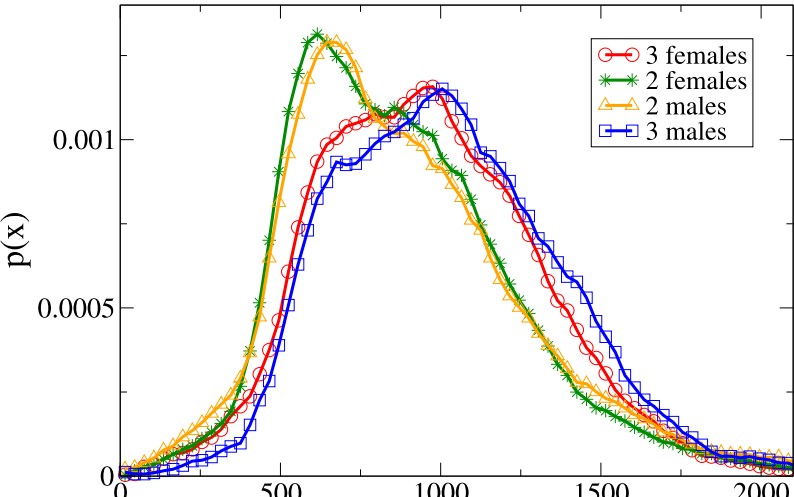

**Fig 13. Pdf of the *x* observable in triads according to gender.** Red and circles: three females. Green and stars: two females. Orange and triangles: two males. Blue and squares: three males.

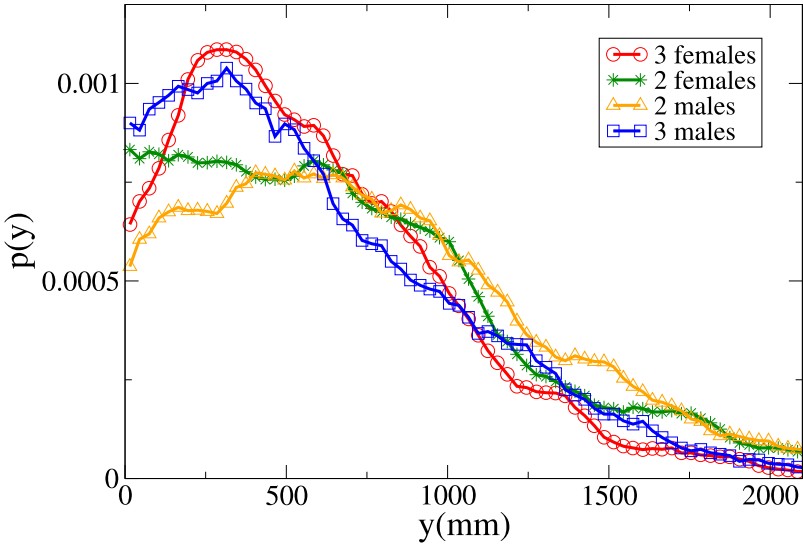

**Fig 14. Pdf of the *y* observable in triads according to gender.** Red and circles: three females. Green and stars: two females. Orange and triangles: two males. Blue and squares: three males.

## The effect of age and height

Purpose and relation are discrete properties of groups. Gender is, strictly speaking, a property of individuals, but may be naturally mapped to a discrete property of the group ("number of females"). On the other hand, age and height are continuous properties of individuals, and for this reason the analysis of their effect on group dynamics is less straightforward (without considering the increased difficulty in coding age). This problem gets obviously more serious as the number of pedestrians in the group grows (because the age may be more diverse). In [46] we decided to analyse the dependence of dynamics on the "minimum age" and "minimum height" of group members, since these allowed us to spot the presence of children, and distinguish the behaviour of families with children from the one of families including only adults. For integrity, we report below the corresponding triadic results, but in this work our analysis of the effect of age and height will be limited to the discrete observables. A statistical analysis of the overall probability distributions concerning minimum age and height is provided in S4 Appendix.

**Age.** Table 7 shows the minimum age dependence of all observables (based on the analysis of 687 triads), while the dyad results from [46] are reported in Table 8. It may be observed, as expected, that groups including children and groups including only elderly people move with a slower velocity. Furthermore, groups with children present the tendency of a less ordered structure (low *x* and high *y*) which is typical of families.

**Height.** Table 9 shows the minimum height dependence of all observables (based on the analysis of 686 triads), while the dyad results from [46] are reported in Table 10. Along with the expected correlation between the behaviour of shorter people with family/children, we can also notice the expected tendency of taller people to walk faster. In dyads (refer to [46]) we had noticed that such a tendency is still present even when accounting for other factors (e.g. taller males walk faster than shorter males).

## The effect of interaction

**Overall effect of interaction on dyads and triads.** Table 11 shows the effect on all observables of the presence of social interaction in dyads, regardless of intrinsic properties. It may be

**Table 7. Observable dependence on minimum age for triads.** Lengths in millimetres, times in seconds.

| Minimum age | $N_g^k$ | V | r | x | y |
|---|---|---|---|---|---|
| 0-7 years | 131 | 973 ± 14 (166) | 621 ± 15 (166) | 881 ± 23 (266) | 723 ± 35 (402) |
| 8-19 years | 270 | 1015 ± 9.2 (151) | 589 ± 8.4 (139) | 902 ± 16 (259) | 685 ± 21 (344) |
| 20-29 years | 121 | 1043 ± 14 (152) | 582 ± 11 (121) | 964 ± 23 (253) | 646 ± 29 (324) |
| 30-39 years | 52 | 1081 ± 26 (185) | 639 ± 22 (159) | 972 ± 44 (316) | 774 ± 62 (445) |
| 40-49 years | 72 | 1094 ± 24 (201) | 625 ± 16 (138) | 1006 ± 34 (290) | 711 ± 43 (365) |
| 50-59 years | 26 | 1080 ± 34 (172) | 603 ± 25 (127) | 1035 ± 51 (261) | 587 ± 57 (291) |
| 60-69 years | 4 | 1038 ± 130 (267) | 709 ± 66 (132) | 1276 ± 160 (322) | 676 ± 270 (547) |
| ≥ 70 years | 11 | 932 ± 41 (137) | 698 ± 61 (202) | 899 ± 110 (358) | 943 ± 95 (315) |
| $F_{7,679}$ | | 5.78 | 2.67 | 3.78 | 1.88 |
| $p$ | | $1.62 \cdot 10^{-6}$ | 0.00993 | 0.000496 | 0.0708 |
| $R^2$ | | 0.0562 | 0.0268 | 0.0375 | 0.019 |
| $\delta$ | | 0.833 | 1.04 | 1.48 | 1.2 |

**Table 8. Observable dependence on minimum age for dyads.** Lengths in millimetres, times in seconds.

| Minimum age | $N_g^k$ | V | r | x | y |
|---|---|---|---|---|---|
| 0-9 years | 31 | 1143 ± 42 (235) | 995 ± 69 (383) | 529 ± 34 (189) | 701 ± 87 (485) |
| 10-19 years | 63 | 1158 ± 33 (259) | 791 ± 33 (259) | 624 ± 19 (148) | 359 ± 40 (320) |
| 20-29 years | 364 | 1181 ± 9.1 (173) | 793 ± 11 (218) | 668 ± 8.1 (154) | 307 ± 14 (264) |
| 30-39 years | 292 | 1204 ± 12 (202) | 836 ± 14 (238) | 673 ± 10 (176) | 364 ± 18 (307) |
| 40-49 years | 149 | 1181 ± 14 (176) | 841 ± 18 (224) | 664 ± 13 (158) | 384 ± 26 (311) |
| 50-59 years | 111 | 1164 ± 18 (193) | 825 ± 21 (219) | 649 ± 15 (160) | 378 ± 30 (318) |
| 60-69 years | 67 | 1028 ± 21 (170) | 881 ± 41 (335) | 638 ± 20 (164) | 468 ± 52 (422) |
| ≥ 70 years | 12 | 886 ± 29 (99.8) | 786 ± 79 (275) | 588 ± 19 (66.6) | 385 ± 100 (363) |
| $F_{7,1081}$ | | 10.7 | 3.96 | 4.23 | 8.02 |
| $p$ | | $<10^{-8}$ | 0.000282 | 0.000128 | $<10^{-8}$ |
| $R^2$ | | 0.065 | 0.025 | 0.0267 | 0.0494 |
| $\delta$ | | 1.6 | 0.583 | 0.808 | 1.37 |

noticed that interaction has a significant effect on all variables. Interacting dyads are slower, walk closer (smaller $r$) with a smaller abreast distance and group depth. Table 12 shows the same result for triads. We have again statistical significance for differences in all observables, although reduced for $V$ and in particular for $x$. Interacting triads are still slower (the effect size

**Table 9. Observable dependence on minimum height for triads.** Lengths in millimetres, times in seconds.

| Minimum height | $N_g^k$ | V | r | x | y |
|---|---|---|---|---|---|
| < 140 cm | 134 | 1003 ± 15 (176) | 643 ± 14 (163) | 893 ± 25 (293) | 756 ± 33 (378) |
| 140-150 cm | 54 | 1027 ± 21 (153) | 612 ± 24 (175) | 879 ± 40 (293) | 800 ± 60 (437) |
| 150-160 cm | 221 | 995 ± 10 (154) | 585 ± 8.6 (128) | 923 ± 17 (258) | 661 ± 23 (338) |
| 160-170 cm | 220 | 1046 ± 11 (163) | 597 ± 9.3 (139) | 944 ± 17 (248) | 688 ± 24 (361) |
| 170-180 cm | 57 | 1125 ± 25 (188) | 611 ± 19 (147) | 1069 ± 43 (328) | 618 ± 46 (344) |
| $F_{4,681}$ | | 8.62 | 3.61 | 4.97 | 3.2 |
| $p$ | | $8.67 \cdot 10^{-7}$ | 0.0064 | 0.000597 | 0.0128 |
| $R^2$ | | 0.0482 | 0.0207 | 0.0283 | 0.0185 |
| $\delta$ | | 0.808 | 0.411 | 0.609 | 0.463 |

**Table 10. Observable dependence on minimum height for dyads.** Lengths in millimetres, times in seconds.

| Minimum height | $N_g^k$ | V | r | x | y |
|---|---|---|---|---|---|
| < 140 cm | 39 | 1130 ± 34 (211) | 1004 ± 65 (404) | 573 ± 34 (210) | 672 ± 80 (501) |
| 140-150 cm | 39 | 1106 ± 50 (311) | 875 ± 46 (289) | 619 ± 25 (156) | 469 ± 64 (403) |
| 150-160 cm | 234 | 1104 ± 13 (197) | 797 ± 16 (246) | 631 ± 8.9 (136) | 360 ± 21 (328) |
| 160-170 cm | 498 | 1169 ± 8.1 (182) | 821 ± 11 (243) | 657 ± 7.7 (172) | 362 ± 14 (311) |
| 170-180 cm | 262 | 1242 ± 11 (173) | 827 ± 12 (197) | 699 ± 9.6 (155) | 321 ± 16 (251) |
| > 180 cm | 17 | 1232 ± 51 (211) | 793 ± 48 (198) | 689 ± 33 (135) | 270 ± 53 (217) |
| $F_{5,1083}$ | | 14.5 | 5.25 | 7.45 | 9.69 |
| $p$ | | $<10^{-8}$ | $9.03 \cdot 10^{-5}$ | $6.9 \cdot 10^{-7}$ | $<10^{-8}$ |
| $R^2$ | | 0.0626 | 0.0237 | 0.0333 | 0.0428 |
| $\delta$ | | 0.744 | 0.591 | 0.773 | 0.922 |

**Table 11. Observable dependence on presence of social interaction in dyads.** Lengths in millimetres, times in seconds.

| Interaction | $N_g^k$ | V | r | x | y |
|---|---|---|---|---|---|
| Interacting | 870 | 1150 ± 6.6 (194) | 795 ± 7.5 (222) | 648 ± 4.9 (144) | 335 ± 9.9 (292) |
| Non-Interacting | 219 | 1251 ± 13 (187) | 947 ± 20 (291) | 695 ± 15 (224) | 485 ± 26 (389) |
| $F_{1,1087}$ | | 48.4 | 72.1 | 14.2 | 40.1 |
| $p$ | | $<10^{-8}$ | $<10^{-8}$ | 0.000174 | $<10^{-8}$ |
| $R^2$ | | 0.0426 | 0.0622 | 0.0129 | 0.0356 |
| $\delta$ | | 0.526 | 0.643 | 0.285 | 0.479 |

**Table 12. Observable dependence on presence of social interaction in triads.** Lengths in millimetres, times in seconds.

| Interaction | $N_g^k$ | V | r | x | y |
|---|---|---|---|---|---|
| Interacting | 536 | 1015 ± 7 (162) | 587 ± 6 (139) | 943 ± 12 (272) | 644 ± 15 (352) |
| Non-Interacting | 151 | 1068 ± 15 (185) | 666 ± 12 (152) | 892 ± 23 (282) | 876 ± 29 (356) |
| $F_{1,685}$ | | 12.1 | 36 | 4.12 | 50.6 |
| $p$ | | 0.000539 | $<10^{-8}$ | 0.0427 | $<10^{-8}$ |
| $R^2$ | | 0.0173 | 0.0499 | 0.00599 | 0.0688 |
| $\delta$ | | 0.321 | 0.553 | 0.187 | 0.657 |

being reduced with respect to dyads) and walk closer. The group depth is considerably reduced by interaction (stronger effect size than dyads). Interestingly the abreast distance $x$ is larger in interacting triads, a result that, albeit with a weaker effect, goes in the opposite direction of the dyadic case.

To understand the difference between the effect of interaction on the spatial structure of dyads and triads we refer to the 2D pdfs for the position of interacting and non-interacting dyads (Fig 15) and interacting and non-interacting triads (Fig 16). Furthermore, the 1D pdfs for the interacting and non-interacting dyads are found in Figs 17, 18, 19, 20, for, respectively, the $V$, $r$, $x$ and $y$ observables, while the corresponding triad observable pdfs are found in Figs 21, 22, 23, 24. In both dyads and triads lack of interaction loosens the group spatial structure. As discussed in [3], dyads have a tendency to walk abreast, and triads in a V formation. Both formations, in order to facilitate interaction, are characterised by having $x > y$, i.e. occupy a larger portion of space in the direction orthogonal to the one of motion. Furthermore, we may

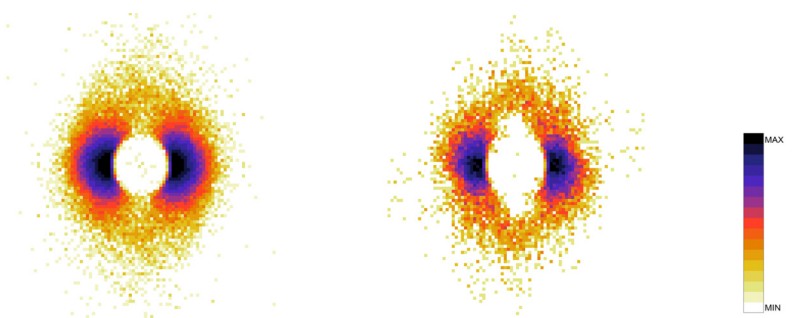

**Fig 15. 2D pdf for the position of dyads.** Left: interacting. Right: non-interacting.

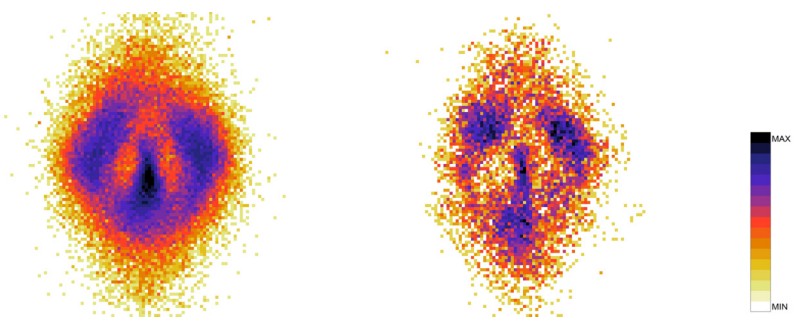

**Fig 16. 2D pdf for the position of triads.** Left: interacting. Right: non-interacting.

expect such spatial structures to be characterised by having $x > 500$ mm, i.e. we expect the width of the group to be larger than human shoulders. Figs 19 and 23 show that the probability of having $x < 500$ mm is increased in both non-interacting dyads and triads. Nevertheless, while for non-interacting triads also the peak and tail of the $x$ distribution are displaced to the

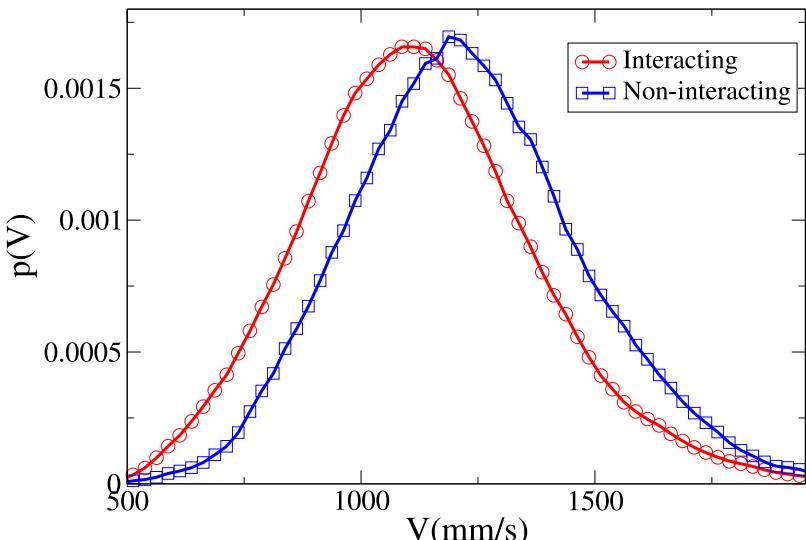

**Fig 17. Pdf of the $V$ observable in dyads.** Red and circles: interacting. Blue and squares: non-interacting.

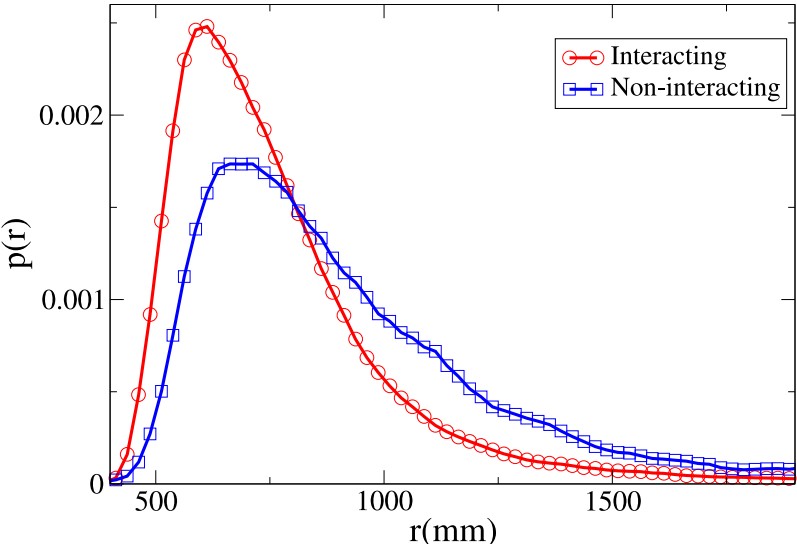

**Fig 18. Pdf of the *r* observable in dyads.** Red and circles: interacting. Blue and squares: non-interacting.

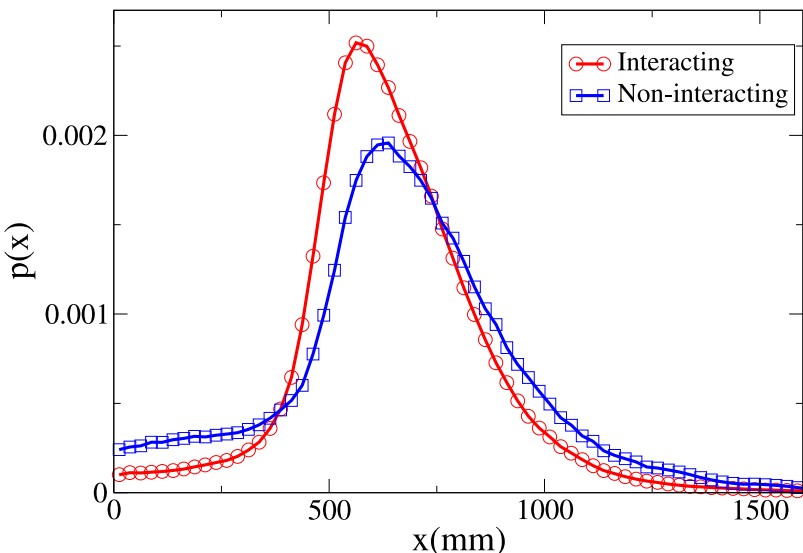

**Fig 19. Pdf of the *x* observable in dyads.** Red and circles: interacting. Blue and squares: non-interacting.

left, for dyads we have the opposite effect (larger peak value and fatter tail for non-interacting dyads). As a result, lack of interaction leads to an *increase* in the average value of *x*.

**Interaction and intrinsic properties.** In S5 Appendix we analyse the interplay between interaction and intrinsic properties such as relation and gender. The main patterns in the observable dependence on gender and relation are present both in interacting and non-interacting dyads and triads; and that for a fixed intrinsic property (e.g. for groups composed of colleagues) there are in general statistically significant differences between interacting and non-interacting triads. We also analyse whether the tendency of non-interacting triads to have a lower *x* extension is affected by density, and we find that although the effect is indeed stronger at higher density, it is present in any density range.

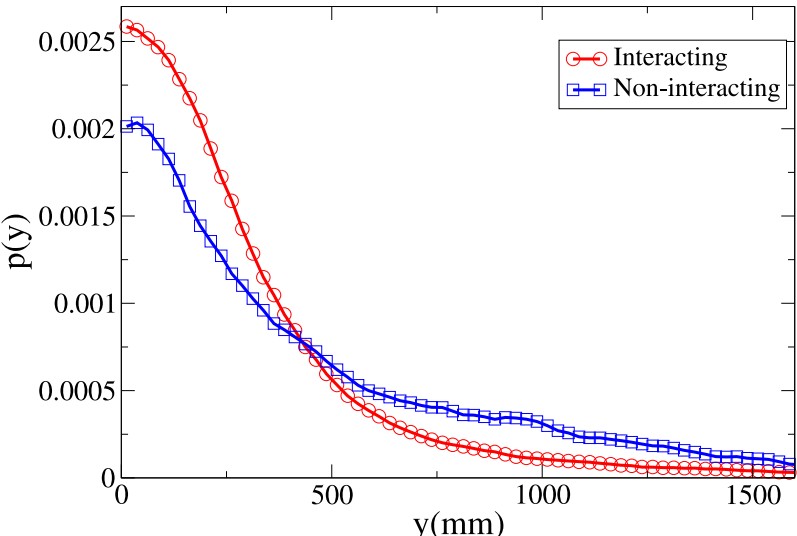

**Fig 20. Pdf of the *y* observable in dyads.** Red and circles: interacting. Blue and squares: non-interacting.

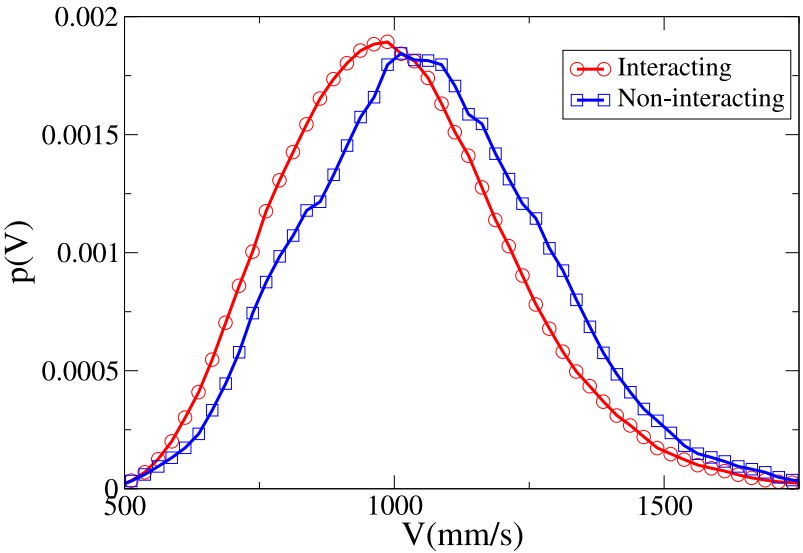

**Fig 21. Pdf of the *V* observable in triads.** Red and circles: interacting. Blue and squares: non-interacting.

## Discussion and conclusion

We analysed how intrinsic properties of moving pedestrian triads, such as their purpose, their personal relation, their gender, age and height, affect their walking dynamics. We have verified a strong parallel between the effect of intrinsic properties on triads and the one on dyads, that we had analysed in a previous work. Work-oriented pedestrians are faster and walk at a larger distance between them than leisure-oriented ones, although leisure-oriented ones move in a less ordered way. Work-oriented triads overlap with the "colleagues" category, so that the differences above were present also when colleagues were contrasted with friends and families. The similarity between friend behaviour and colleague behaviour is larger than the one

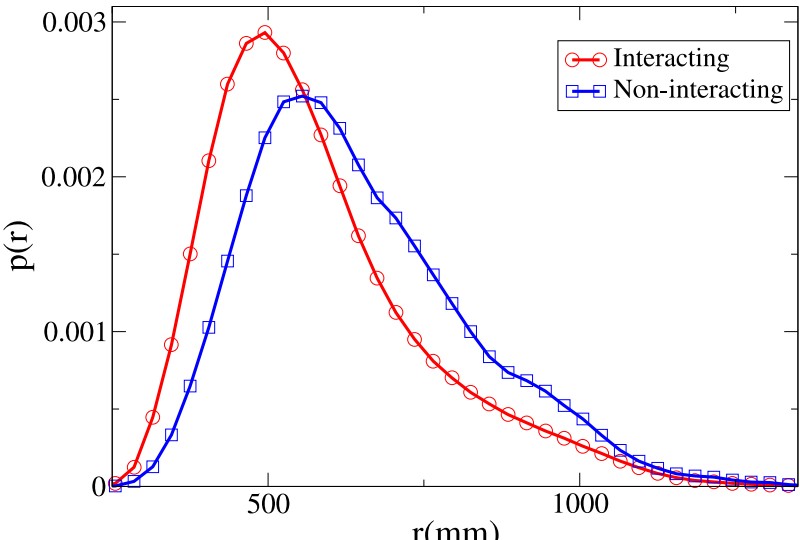

**Fig 22. Pdf of the *r* observable in triads.** Red and circles: interacting. Blue and squares: non-interacting.

between family behaviour and colleague behaviour (friends are faster and more ordered). We also found that all male triads walk faster than triads including females, that males keep a larger distance than females, and that same gender groups are more ordered than mixed ones, an effect probably due to the presence of families and children. Although the analysis on age and height was more difficult than in the work on dyads, we found evidence that triads composed of elderly people and those including children walk at a slower pace. Groups including children are found to move in a less ordered fashion. Finally, we found that groups composed of tall people walk faster.

For the dependence on relation and gender, we explicitly verified whether the above results hold also when other properties, including crowd density, are kept fixed. The effect of relation

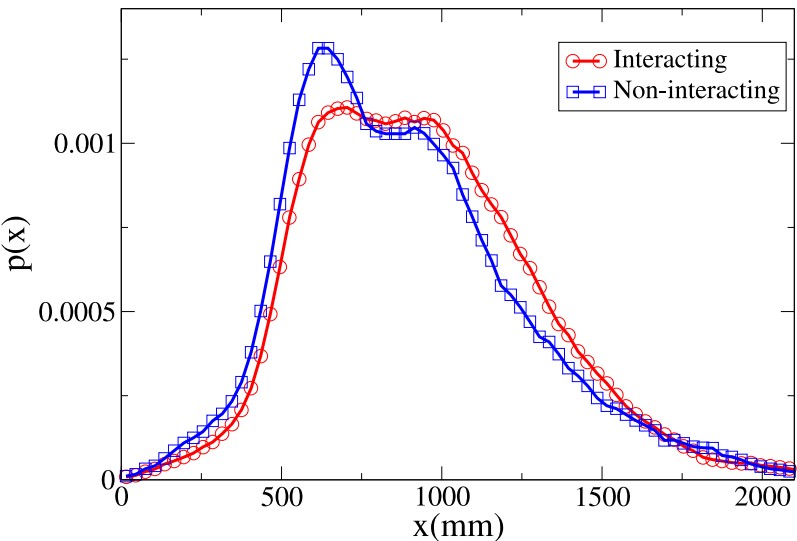

**Fig 23. Pdf of the *x* observable in triads.** Red and circles: interacting. Blue and squares: non-interacting.

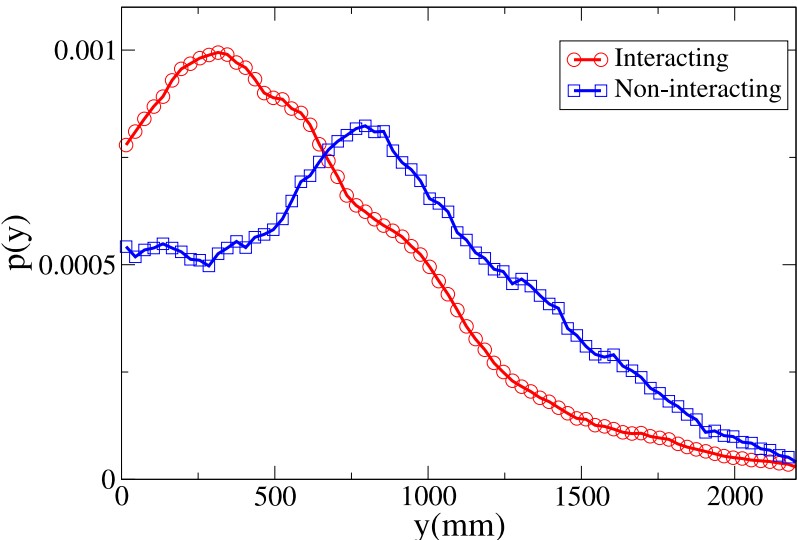

**Fig 24. Pdf of the *y* observable in triads.** Red and circles: interacting. Blue and squares: non-interacting.

is fundamentally the same irrespective of other extrinsic and intrinsic properties. The patterns found for the gender composition of groups resulted to be stable but often diminished when other properties were kept fixed. For instance, although male triads are in general faster than female ones, the difference is reduced if we compare only triads in the colleague relation. Similarly, the apparently more ordered structure of all male triads is mostly due to the small number of families.

We also analysed, for both dyads and triads, the effect of explicit social interaction at the time of observation between all members of the group on its dynamical properties. We found that the effects on group (dyad and triad) dynamics due to intrinsic properties are present regardless of social interaction. Nevertheless, social interaction has a statistically significant influence on dyad and triad dynamics, which is mainly expressed as a tendency to walk in a more ordered fashion when interacting. Interestingly, this has an opposite effect on the space occupied by non-interacting dyads and triads in the direction orthogonal to that of movement, since loss of structure makes dyads larger, but causes triads to lose their characteristic V formation and walk in a line (i.e., occupying more space in the direction of movement but less space in the orthogonal one).

We believe that our findings may deepen our understanding of crowd dynamics and the reliability of simulations. Although our findings inherently apply to the kind of environment and conditions that we used to collect our data (a shopping mall under normal working day and weekend conditions, with density levels from low to moderate), arguably a better understanding of different group behaviour in these conditions may work as a guiding light also in the analysis of more general settings. The different levels of "attachment" between group members depending on relation, for example, could have an important effect on evacuation times, although obviously the values reported in this paper are not supposed to be trivially generalised outside of their applicable range. Similarly, it may be expected that in high density settings pedestrian groups may stop social interaction, which as shown in this work has an important effect on their dynamics. Similarly, interaction level may be modified (decreased or increased, according to different possible scenarios) in emergency situations.

Possible applications beyond pedestrian simulation may regard robot and mobile vehicle navigation [66–69], and pedestrian detection [44, 52, 70].

## Supporting information

**S1 Data Set. Interacting dyads trajectories and coding data set.**
(ZIP)

**S2 Data Set. Non-interacting dyads trajectories and coding data set.**
(ZIP)

**S3 Data Set. Interacting triads trajectories and coding data set.**
(ZIP)

**S4 Data Set. Non-interacting triads trajectories and coding data set.**
(ZIP)

**S1 Annex. Information concerning coding criteria.**
(ZIP)

**S1 Appendix. Coder reliability.**
(PDF)

**S2 Appendix. Statistical analysis of observables.**
(PDF)

**S3 Appendix. Accounting for other effects.**
(PDF)

**S4 Appendix. Statistical analysis of overall probability distributions.**
(PDF)

**S5 Appendix. Interaction and intrinsic properties.**
(PDF)

## Acknowledgments

We would like to thank Dr. Dražen Brščić for planning and carrying out the experiments; and providing the tracking outcomes. We would like to thank Ms. Kanako Tomita for helping the annotation process. This work was supported by JST CREST Grant Number JPMJCR17A2, Japan. This work was supported by JSPS KAKENHI Grant Number J18K18168, Japan.

## Author Contributions

**Conceptualization:** Francesco Zanlungo.

**Data curation:** Francesco Zanlungo.

**Formal analysis:** Francesco Zanlungo, Zeynep Yücel.

**Funding acquisition:** Takayuki Kanda.

**Investigation:** Francesco Zanlungo.

**Methodology:** Francesco Zanlungo.

**Project administration:** Takayuki Kanda.

**Software:** Francesco Zanlungo.

**Validation:** Francesco Zanlungo.

**Visualization:** Francesco Zanlungo.

**Writing – original draft:** Francesco Zanlungo, Zeynep Yücel.

**Writing – review & editing:** Francesco Zanlungo, Zeynep Yücel, Takayuki Kanda.

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
