## [Decision Letter · Decision Letter 0]

26 Sep 2019

PONE-D-19-21011

Intrinsic group behaviour II: on triads and interaction

PLOS ONE

Dear Dr. Zanlungo,

Thank you for submitting your manuscript to PLOS ONE. After careful consideration, we feel that it has merit but does not fully meet PLOS ONE’s publication criteria as it currently stands. Therefore, we invite you to submit a revised version of the manuscript that addresses the points raised during the review process.

Both Reviewers believe that this manuscript is of potential interest for PLoS one, after minor revisions. Although I do not necessarily agree with every single comment of Reviewer #1, I believe that s/he raises important interpretation issues, and the authors should expand the discussion of their results, following the Reviewer #1's suggestions. The suggestions of Reviewer #2 should be straightforward to implement.

We would appreciate receiving your revised manuscript by Nov 10 2019 11:59PM. To enhance the reproducibility of your results, we recommend that if applicable you deposit your laboratory protocols in protocols.io, where a protocol can be assigned its own identifier (DOI) such that it can be cited independently in the future. For instructions see: http://journals.plos.org/plosone/s/submission-guidelines#loc-laboratory-protocols

We look forward to receiving your revised manuscript.

Kind regards,

Roberta Sinatra

Academic Editor

PLOS ONE

Journal Requirements:

2. Please provide additional details regarding obtaining participant consent from the coders who participated in your study. In the ethics statement in the Methods and online submission information, please ensure that you have specified (1) whether consent was informed and (2) what type you obtained (for instance, written or verbal, and if verbal, how it was documented and witnessed).

The authors declare that no competing interests exist. Authors FZ, ZY, and TK are affiliated to a private company, Advanced Telecommunications Research Institute  International (ATR), respectively, as part-time contract researcher (FZ),  visiting researcher (ZY), and visiting group leader (TK). Nevertheless, their activity in ATR concerning this research work is related to the development of the government and university funded projects stated in the financial disclosure, and not to the development

of commercial products or other economical interest of ATR. The authors state that no work—acknowledging all financial support patent application is going to be pursued based on the research presented on this work, and that the results of this work are not going to be used to promote any kind of economical interest.

4. We noted in your submission details that a portion of your manuscript may have been presented or published elsewhere: 

As stated in the cover letter, this work is a follow-up to our 2017 PloS One publication.

Zanlungo F, et al.,

"Intrinsic group behaviour: Dependence of pedestrian dyad dynamics on principal social and personal features",

PLoS One, 12, 11, e0187253 (Public Library of Science, 2017)

For the sake of integrity, we copied some tables concerning the behaviour of dyads from the aforementioned publication, to which we compare the new results concerning the behaviour of triads. When we present material from the 2017 publication, we always clearly state this fact in the text.

Reviewers' comments:

Reviewer's Responses to Questions

**Comments to the Author**

1. Is the manuscript technically sound, and do the data support the conclusions?

Reviewer #1: Yes

Reviewer #2: Yes

2. Has the statistical analysis been performed appropriately and rigorously? 

Reviewer #1: Yes

Reviewer #2: Yes

3. Have the authors made all data underlying the findings in their manuscript fully available?

Reviewer #1: Yes

Reviewer #2: Yes

4. Is the manuscript presented in an intelligible fashion and written in standard English?

Reviewer #1: Yes

Reviewer #2: Yes

5. Review Comments to the Author

Reviewer #1: Title:

In its current form it is too broad and not informative enough. It should be more specific like the title of the previous work on dyads. (i.e.: group behavior in the physical space / intrinsic properties of moving pedestrian triads, etc.) It should express that the present study is about how individual and relational features in triads influence their behavior in the physical space.

The manuscript is poorly placed in context, especially when it comes to the scientific contribution. The study claims that groups’ presence influences the dynamics of the crowd, moreover, it is assumed that the behavior of these groups may have an impact on evacuation time and the planning of buildings. However, it is not clear or even mentioned on the level of assumptions, how the findings of this research contribute to understand how the presence and dynamics of the groups under observation influence the dynamics of the crowd; and how it could contribute to evacuation plans in emergency situations. As a consequence, the purpose and the scientific contribution of the study is unclear.

Line 116: The behaviour of walking groups depends also on its intrinsic properties. Age,

gender and height are known to affect walking speed (as observed in studies with

subjects [38]).” The listed properties are not possessed by the group but by the individual group members. At other points of the manuscript, these features are also referred to as group properties incorrectly. Age composition, for example, would be a better choice.

Line 69: “..data set and described how spatial structure, size and velocity of dyads (two people groups) depend on intrinsic properties of groups, and more specifically on:...”

The “size of dyads“ is incomprehensible in this way.

Also, the term ‘size’ / ‘group size’ is used in an ambiguous way several times (i.e.: line 95) in the introduction, since its exact meaning in the given context is defined much later in the text. Using the term before conceptualization is ambiguous and misleading.

Line 81: “Nevertheless, they provided a useful and quantitative insight into how intrinsic

features affect dyadic behaviour.” The manuscript does not say anything about how and why this insight is useful.

Did human coders receive any training, written guidance including explicit rules for coding? If so, it is supposed to be attached to the annexes.

Result section:

Line 301 – Based on the analysis, triads are always slower than dyads for any values of all intrinsic or extrinsic properties. However, this result is hardly explainable with these properties. A (side)note about the potential effect of group size and coordination strategies should be added or noted as an important aspect that is not detailed in the present study.

It would be useful to reflect on the results briefly, not only show the table and merely describe what we can see there, but add a thought / assumption or just play with the idea of what could explain these findings (i.e.: Why can we see high values for group depth in families?)

Discussion:

The term “intrinsic properties” is put into quotation marks in the abstract and discussion section while it is present without quotation marks in other parts of the manuscript.

Also, in the discussion part, (line 471) “ordered” groups is used for the first time, and it is unclear what accounts for an ordered group in this context.

The discussion is also lacking of presenting how the findings on triads’ behavior in physical space contribute to a deeper understanding of crowd dynamics and the evacuation of a building. The absence of explanation makes this connection vague and forced.

Typographical and grammatical errors

Line364 – capital D for differences in the beginning of the sentence.

Table 8: ‘Observable dependence on minimum age for dyads’ is in the ‘Height’ section – should be in the ‘Age’ section.

In the ‘Effect of interaction‘ section, the Table 11. and 12. should be swapped for the sake of consistency, as tables on triads are followed by the ones on dyads at every other part of the manuscript.

References

Reference number 10 (He L, Pan J, Wang W, Manocha D. Proxemic group behaviors using reciprocal multi-agent navigation. In: Proceedings of the International Conference on

Robotics and Automation. IEEE; 2016. p. 292–297.) is not used/flagged in the manuscript.

(Most of the used references correspond to the reference list of the authors’ previous work on dyads.)

In general: The manuscript is technically sound. A great amount of ecological data coupled with the achieved high inter-coder reliability is a convincing start for meaningful results. The authors performed an appropriate and rigorous statistical analysis. The data supports the conclusion, and the manuscript is aligned with PLOS Data Policy. It is written in standard English.

At the same time, it is clear that first data analysis took place and then, the framework was constructed around the data, that it is not fortunate to present it such explicitly.

The authors properly describe the tables and figures displaying the results, but there is no attempt for interpretation not even to share a few assumptions/ideas about how we should understand and place these findings to a comprehensive picture about group behavior in physical space. This is missing both from the introduction and the discussion part, as well as, a clear and proper connection of the present findings to the claimed purpose of the study.

The introduction is poorly constructed since several terms are used ambiguously. While the reader gets to the point where these terms such as ‘size’ or ‘relation between members’ are properly conceptualized and operationalized, they probably get tired and confused by the inconsistency. The authors make these terms clear later, but the other way around would be much more appreciated.

They correctly draw attention both to the potential cultural limitations and the weakness of the apparent belonging approach. Since most of the findings are not surprising but supporting common sense, it would be beneficial to enrich the study by framing them in a colorful way, so it is embedded into a comprehensive image about group behavior in the physical space.

Reviewer #2: The authors investigate triads walking dynamics. They study how aspects including age, gender, heights of the group members, together with their relation with each other and the purpose of their trip has an effect on group velocity and distance between group members. The analysis is based on a large-scale dataset collecting trajectories of pedestrians within a large multi-purpose building in Osaka. The article is a follow-up of a previous work on the walking dynamics of dyads that two of the authors published in Physical Review E.

Understanding pedestrian walking dynamics is important to plan buildings and emergency evacuation, and the topic may be of interest for the broad readership of PLOS One. The article is clearly written and the statistical analyses have been performed rigorously. I judge the article is suitable for publication in PLOS One, pending the minor changes below.

As a general comment, while the authors investigate in depth the role played by each individual property of the group, they do not dig into the interplay between different factors. For example, one could test a multilinear regression analysis, and study the relative importance of the different variables (for example via Lindeman, Merenda and Gold)

Below, there are some minor comments:

Typo, line 37: “For this reasons” → “For these reasons”

Sentence lines 133-138: Please cut this sentence into multiple sentences.

Line 132: Please add some more information about how participants were recruited and what do you mean by multi-purpose building.

Line 212: Add references to “Cohen’s kappa and Krippendorf’s alpha”

Line 278: Please define O.

Line 281: Please give a quick definition of the presented metrics (ANOVA p-value, effect size, coefficient of determination).

Table 1 (and all tables): provide more explications in the table captions

6. PLOS authors have the option to publish the peer review history of their article (what does this mean?). If published, this will include your full peer review and any attached files.

Reviewer #1: Yes: Rebeka O. Szabo

Reviewer #2: No

---

## [Author Response · Author response to Decision Letter 0]

14 Oct 2019

See atteched pdf file (rebuttal.pdf)

---

## [Decision Letter · Decision Letter 1]

12 Nov 2019

Intrinsic Group Behaviour II: on the Dependence of Triad Spatial Dynamics on Social and Personal Features; and on the Effect of Social Interaction on Small Group Dynamics

PONE-D-19-21011R1

Dear Dr. Zanlungo,

We are pleased to inform you that your manuscript has been judged scientifically suitable for publication and will be formally accepted for publication once it complies with all outstanding technical requirements.

With kind regards,

Roberta Sinatra

Academic Editor

PLOS ONE

Reviewers' comments:

Reviewer's Responses to Questions

**Comments to the Author**

1. If the authors have adequately addressed your comments raised in a previous round of review and you feel that this manuscript is now acceptable for publication, you may indicate that here to bypass the “Comments to the Author” section, enter your conflict of interest statement in the “Confidential to Editor” section, and submit your "Accept" recommendation.

Reviewer #1: All comments have been addressed

Reviewer #2: All comments have been addressed

2. Is the manuscript technically sound, and do the data support the conclusions?

Reviewer #1: Yes

Reviewer #2: Yes

3. Has the statistical analysis been performed appropriately and rigorously? 

Reviewer #1: Yes

Reviewer #2: Yes

4. Have the authors made all data underlying the findings in their manuscript fully available?

Reviewer #1: Yes

Reviewer #2: Yes

5. Is the manuscript presented in an intelligible fashion and written in standard English?

Reviewer #1: Yes

Reviewer #2: Yes

6. Review Comments to the Author

Reviewer #1: The authors have adequately addressed all the comments. I find the manuscript technically sound, clear and coherent in its current form. In my opinion, it is suitable for publication.

Reviewer #2: I would like to congratulate the authors for the revision work. The authors have thoroughly addressed the reviewers comments, by expanding the introduction and clarifying some previously unclear points.

7. PLOS authors have the option to publish the peer review history of their article (what does this mean?). If published, this will include your full peer review and any attached files.

Reviewer #1: No

Reviewer #2: No

---

## [Editor Report · Acceptance letter]

15 Nov 2019

PONE-D-19-21011R1 

Intrinsic Group Behaviour II: on the Dependence of Triad Spatial Dynamics on Social and Personal Features; and on the Effect of Social Interaction on Small Group Dynamics 

Dear Dr. Zanlungo:

I am pleased to inform you that your manuscript has been deemed suitable for publication in PLOS ONE. Congratulations! Your manuscript is now with our production department. 

With kind regards,

on behalf of

Prof. Roberta Sinatra 

Academic Editor

PLOS ONE